# Mobile phone use is associated with higher smallholder agricultural productivity in Tanzania, East Africa

Amy Quandt[1]*, Jonathan D. Salerno[2], Jason C. Neff[3], Timothy D. Baird[4], Jeffrey E. Herrick[5], J. Terrence McCabe[6], Emilie Xu[7], Joel Hartter[3]

1 Department of Geography, San Diego State University, San Diego, California, United States of America, 2 Department of Human Dimensions of Natural Resources, Colorado State University, Fort Collins, Colorado, United States of America, 3 Environmental Studies Program, University of Colorado Boulder, Boulder, Colorado, United States of America, 4 Department of Geography, Virginia Polytechnic Institute and State University, Blacksburg, Virginia, United States of America, 5 Jornada Experimental Range, United States Department of Agriculture – Agricultural Research Service, Las Cruces, New Mexico, United States of America, 6 Department of Anthropology, University of Colorado Boulder, Boulder, Colorado, United States of America, 7 Fairview High School, Boulder, Colorado, United States of America

* aquandt@sdsu.edu

**Data Availability Statement:** Data cannot be shared publicly due to IRB protocols and standards of privacy and confidentiality. Data are available by request from the University of Colorado Boulder

## Abstract

Mobile phone use is increasing in Sub-Saharan Africa, spurring a growing focus on mobile phones as tools to increase agricultural yields and incomes on smallholder farms. However, the research to date on this topic is mixed, with studies finding both positive and neutral associations between phones and yields. In this paper we examine perceptions about the impacts of mobile phones on agricultural productivity, and the relationships between mobile phone use and agricultural yield. We do so by fitting multilevel statistical models to data from farmer-phone owners ($n = 179$) in 4 rural communities in Tanzania, controlling for site and demographic factors. Results show a positive association between mobile phone use for agricultural activities and reported maize yields. Further, many farmers report that mobile phone use increases agricultural profits (67% of respondents) and decreases the costs (50%) and time investments (47%) of farming. Our findings suggest that there are opportunities to target policy interventions at increasing phone use for agricultural activities in ways that facilitate access to timely, actionable information to support farmer decision making.

## Introduction

The rapid diffusion of mobile phones in the Global South has increased information flow, reduced telecommunication costs, and led to novel strategies for economic development [1, 2]. In a study of 120 developing countries, growth in mobile phone penetration coincided with economic growth [3]. Mobile phones have impacted the lives of hundreds of millions, particularly in areas with poor access to landline telephones due to a lack of infrastructure or electricity. In Sub-Saharan Africa, phones are increasingly used to provide a host of services and information across the financial, energy, and agronomic sectors. Furthermore, the ubiquity of

Research and Innovation Office (rio@colorado.edu) in conjunction with the corresponding author (contact via aquandt@sdsu.edu) for researchers who meet the criteria for access to confidential data.

**Funding:** This work was supported by the University of Colorado's Research & Innovation Seed Grant Program.

**Competing interests:** The authors have declared that no competing interests exist.

mobile phones throughout sub-Saharan Africa offers new opportunities for rural households to realize a broader set of livelihood and development goals [4]. Information and communication technologies (ICTs), including mobile phones, have been shown to help reduce poverty in sub-Saharan Africa by strengthening and expanding social networks, cutting down on travel costs, maximizing the outcomes of necessary journeys, managing human-wildlife conflict, conducting business and financial transactions, and increasing the efficiency of livelihood activities [5–7]. For example, in Kenya, access to mobile money services was found to reduce extreme poverty in female-headed households by 22% [8] and more generally to have a positive impact on agricultural household income [9]. The benefits of ICTs have been well documented, but it is also important to note that these technologies can also exacerbate existing power imbalances and inequalities [10].

Sub-Saharan Africa is the fastest growing and second largest mobile market in the world [11]. By the end of 2017 the unique mobile subscriber penetration rate stood at 44% [12]. It is projected that the future growth of mobile phone use will be concentrated in rural areas and with a younger demographic, with approximately 300 million additional people becoming mobile phone subscribers by 2025 [12]. Smartphone connections particularly are expected to increase from 34% of connections in 2017 to 67% of connections by 2025 due to the growth of cheaper devices [12]. Electricity, particularly in rural areas, is a current barrier to mobile phone adoption [13]; however, increased access to electricity, cheaper phones, and lower costs of airtime and data continue to fuel growth [11].

The intersection of the agricultural economy and the expanding use of mobile phones has led to policy innovations related to phones for a range of agricultural services including the connection of farmers to buyers, the provisioning of inputs for farming, and the formal and informal exchange of agricultural information and recommendations [4]. Agriculture is the dominant income-generating activity in rural Sub-Saharan Africa where nearly 9 in 10 households generate income from crop production, and where non-agricultural income generation lags behind that of other developing regions [14]. Despite growing diversification of household incomes, agriculture remains the primary livelihood activity in rural areas and a focal point for economic development policies and interventions. ICTs for agricultural development initiatives are growing in number, with over 140 such initiatives reported globally in 2015 [4]. There is also a growing use of mobile phones for agricultural extension and outreach throughout sub-Saharan Africa, where agricultural extension agents must provide education, advice, and services to farmers across large geographic areas, and have little access to equipment and knowledge platforms [13, 15].

Over the past decade, the spread of these technologies has led to macro-scale improvements in agricultural market performance in developing economies [16] but with more mixed impacts locally with individual households and farmers [17]. Despite phones' potential in the agricultural sector, there is mixed evidence on the relationship between the use of mobile phones and improved yields with evidence for positive [18–20] and neutral [21, 22] associations between ICTs and yields. Diverse, reported impacts may stem from variation in structural issues such as access to markets, transportation infrastructure, and ICTs across local contexts [23]. The diversity of conclusions may also be due in part to differences in methodologies, phone use measurements, and the many factors that must be considered in the analyses.

Many studies of mobile phone use and agricultural productivity are econometric [4], focused on outcomes and impact evaluations [24], concerned with specific agricultural services/projects [4], and based on large, national datasets [25]. However, fewer studies have examined farmers' perceptions of ICTs and the value they hold for agriculture despite the critical role perceptions play in the adoption of innovations [26]. Often, the perception of the characteristics of an innovation, including its benefits, drives its rate of adoption [26]. This current

study is unique because it focuses on the perceptions of mobile phones specifically for agricultural activities rather than simply phones themselves.

Our study addresses this gap with a survey-based approach to examine farmers' perceptions of mobile phones and agricultural development in 4 rural villages in Iringa Region, Tanzania, where agriculture and fishing are the main sources of household income, 85% of the population had at least a primary education in 2017, nearly two-thirds lived in poverty, and 65% of individuals and 80% of households owned a mobile phone respectively [27]. Here, we use the term "perceptions" to refer to respondents' own perceived behaviors, as opposed to researchers' observations of research subjects' behaviors. Our study provides a unique contribution to ICT4D scholarship by focusing on farmers' perceptions of mobile phones and their specific uses for agricultural practices and effects associations with productivity.

This research addresses 3 main questions: (RQ1) What perceptions do farmers (male and female) have about the impact of mobile phones on agricultural productivity? (RQ2) What is the relationship between generalized mobile phone use and self-reported agricultural yield? and, (RQ3) What is the relationship between mobile phone use specifically for agricultural purposes and self-reported agricultural yield?

Lastly, this research seeks to inform food and agriculture-related policies that affect the use of technological innovations to improve food systems and agricultural productivity. Our study provides further evidence that mobile phone use can be linked to farm activities. This research is at the nexus of food, agriculture, and technology. Surprisingly, there are few existing formal international, national, or local policies addressing the use of mobile phones in the agricultural sector despite their dynamic and emerging nature. Thus, an overarching goal of this research is to inform government and NGOs policies and action plans aimed at improving agricultural productivity.

## Methods

Permission to conduct human subjects research was granted by the Tanzania Commission for Science and Technology (Research Permit No. 2017-250-NA-2017-166), and the University of Colorado Institutional Review Board (protocol # 17–0042).

### Study area

Iringa Region provides an excellent setting to examine these issues. Mobile-phone use has grown steadily in this area and smallholder agriculture is widespread. Furthermore, Tanzania is one of the eight sub-Saharan African markets predicted to contribute more than a third of new mobile subscribers globally between 2016 and 2021 [28]. Additionally, the agricultural sector in Tanzania accounts for more than 45% of the country's GDP, 65% of the export earnings, and engages 80% of the workforce [13]. The study was conducted in the villages of Kibena, Lyamgungwe, Malagosi, and Mgama, located in the Iringa Rural District of Iringa Region in southern Tanzania (Fig 1). Within this district, which had a population of approximately 270,000 in 2017, a government funded assessment reported 53% of residents engage in farming or fishing as their main economic activities, 83% had at least a primary education, and 64% of adults and 90% of households owned a mobile phone [27]. Our study villages, which are overwhelmingly agricultural, were selected because they are ethnically and economically similar, though they differ in population, area, level of development, and distance to a main road. Kibena is the most urban, as it is located on the major highway connecting Tanzania to Zambia. Mgama is located along a well-maintained murram road (hard-packed soil) a few kilometers off the main highway. Lyumgungwe and Malagosi are markedly more rural. They are located along poorly-maintained roads, and are not connected to the electrical grid. The

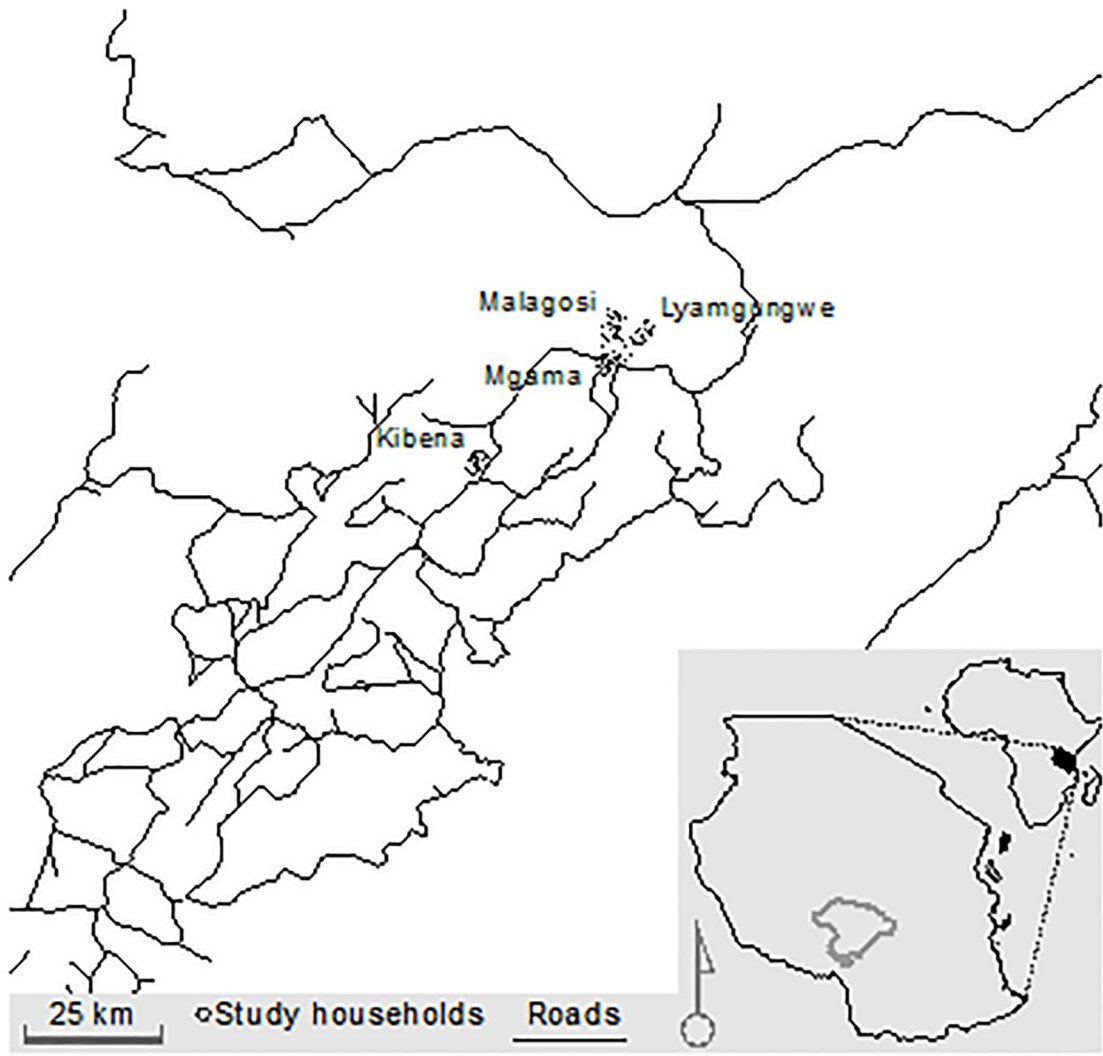

**Fig 1. Study communities in Iringa Region, Tanzania.**

major ethnic group represented in the study villages is Hehe with Bena as the prominent minority group, both of whom are mainly farmers, but also keep cattle and goats. Maize is the primary staple crop in all 4 villages. The majority of agriculture is rain fed. Annual precipitation averages 680mm, and the rains typically begin in late November or early December, and last through April. At least one of the 4 major mobile networks (Tigo, Vodacom, Airtel, and Halotel) is available in each of the 4 villages.

## Data collection

We first conducted qualitative focus group discussions in 3 villages (Nyamihuu, Mapogoro, and Lupalama) to collect basic information about phone use and agriculture in the region. Male, female, and youth focus groups were conducted separately. Villages were selected due to their similarities with the villages that participated in the household survey, including approximate degrees of intra-household economic diversity, various levels of urbanization and/or development, and broad reliance on agriculture. The villages selected for focus groups were

different than for the survey in order to avoid having survey respondents who also participated in the focus groups biasing their answers. The major goal of the focus groups was to inform the survey design, helping to create a contextually-specific survey instrument. Primary data from the focus groups are not presented in our analysis. Focus group discussion notes were qualitatively analyzed and discussed within the research team to find recurring and important themes and ideas about agriculture and mobile phone use that were then integrated into the survey instrument.

A total of 279 surveys were conducted in July 2017, which roughly corresponds to the timing of the maize harvest in the region. Informed consent was granted by respondents prior to surveys. The timing of the surveys is important, and, in the case of this research, conducting surveys during/after the maize harvest proved beneficial because yield and agricultural information was fresh in the minds of the respondents. We implemented a balanced, stratified random sample, with each of the 4 villages as strata; a goal sample of 40 household compounds per village was informed by previous research implementing similar inferential methods [29–31], and set at a level substantially higher than a recent government sponsored economic assessment in the region [27]. Households in each village were selected randomly from the village register, or roster (160 household compounds total across all 4 villages). At each household, our goal was to interview a male and female household member who engages in farming (which was determined before the survey took place), preferably the male and female household heads, or, if absent, another adult household member. On occasion, when an adult (18 or older) of 1 gender was not available, only 1 interview was conducted. Surveys were conducted by enumerators in Kiswahili, the official language of Tanzania, because all survey respondents were fluent in Kiswahili.

Enumerators first underwent a 3-day training before starting data collection, which was led by the lead author in Kiswahili. The enumerators assisted in the translation of the survey from English to Kiswahili, and conducted a practice survey before beginning survey work. Enumerators worked in teams of 2, 1 male and 1 female, with each enumerator only interviewing respondents of their same gender. Using same gender enumerators is a common practice in rural, developing areas where education is relatively low and gender roles can be hierarchical [32]. Groups of enumerators were assisted by local residents who helped to locate the randomly-selected households and provided an introduction to the household members of the research topics on behalf of the research team.

The survey included questions about demographics, phone ownership and use, social networks, and agricultural practices and productivity. Respondents were asked about their own individual behavior, perceptions, and agricultural activities. All respondents were asked the same questions because the goal of the survey was not necessarily to differentiate between groups of people, but instead to understand the overall generalized importance of mobile phone use on agricultural productivity in the study area. Furthermore, we focused the survey on maize, as it is the staple crop in the area, but also recorded other crops planted. Survey responses were based on the respondents' perceptions and observations.

The level and type of phone ownership were determined based on 4 factors: (1) If there was a phone owned by any member of the household; (2) If the respondent personally owned any phone; (3) If they owned an internet capable phone; and (4) If they owned a smartphone. Smartphones and internet capable phones were differentiated for this research because the later are similar to non-smartphone cell phones but have the ability to access the internet (e.g., email, Facebook, etc.) through a simple interface and small screen, lacking the ability to download apps and the touch-based interface of smartphones. Other measures of phone use used in the analysis are described below.

## Data analysis

To examine farmers' perceptions of the impacts of mobile phone technologies on agricultural productivity, we calculated simple descriptive statistics of the survey data (RQ1). To test for associations between general phone use and self-reported agricultural yield (RQ2), and also for associations between phone use specifically for agricultural purposes and self-reported agricultural yield (RQ3), we estimated separate regression models (see below). General maize yields were reported as number of 65-kg sacks per hectare of grain (not on the cob) in recent good years. These values were converted into tons per hectare for presentation and analysis. It is important to note that the weight of maize sacks can vary throughout Tanzania. Long-term residents of Iringa Region told us that they generally use 65 kg as the standard weight in Iringa Region. However, this should be considered an approximation.

RQ2 and RQ3 were operationalized into 2 separate multilevel linear statistical models predicting self-reported maize yield in a good year: a *general phone use* model (RQ2) and *phone for agriculture* model (RQ3). The decision to use multilevel models (i.e., varying effects, random effects, hierarchical models) was informed *a priori* by the nested structure of our data, with respondents in households in communities, and we expect variance to be correlated within these groups [33]. Our choice for multilevel models is supported in 3 ways. First, these models make more accurate estimates than ordinary linear models when data are clustered or share similarities by group, as is the case with multiple observations from the same individual, location, or time period. Second, multilevel models better account for potential imbalances in the sample. Third, multilevel models are also more appropriate when variation within and between groups in the data (i.e., group-level effects) are relevant to research questions [see [34], p. 356 for further discussion]. We do, however, test this assumption by estimating alternative specifications with fixed effect dummy variables for village for both the *general phone use* and *phone for agriculture* models. These are not as parsimonious as the reported models including varying effects for village, as determined by information criterion. (dAICc = 7 and 8, respectively) [35].

Our outcome variable, self-reported maize yield, was continuous and approximated a log-normal distribution, so we fitted Gaussian models. The 2 research questions were specified into 2 separate models conforming to the following structure:

$$y_i \sim \text{Normal}(\mu_i, \sigma)$$

$$\mu_i = \alpha + \beta \ x_{i,h,v} + H + V,$$

where the log of self-reported maize yield $y$ by respondent $i$ has a Gaussian distribution with mean $\mu_i$ and standard deviation $\sigma$. The linear predictor, $\mu_i$, is specified as a function of the grand intercept $\alpha$, a vector of $\beta$ parameters and associated predictor variables $x$ observed in individual $i$, household $h$, and village $v$. For the general phone use model (RQ2), this vector includes the focal variables *total number of contacts*, *number of SMS messages sent and received in the past 24 hours*, and *number of calls made and received in the last 24 hours* plus controls. For the phone for agriculture model (RQ3), this vector includes a single synthetic focal variable measuring phone use specifically for agricultural purposes derived from a set of survey questions, plus controls. These fixed effect variables are detailed just below. Varying intercept effects (i.e., random effects) $H$ and $V$ are understood as the household- and village-level adjustments, respectively, to the linear predictor.

The general phone use model includes 3 focal variables to evaluate the question of whether or not mobile phone use generally was associated with self-reported maize yield outcomes (RQ3). Because "phone use" can be described in multiple ways and be difficult to measure

accurately across various seasons and over long periods of time, we selected 3 proxy variables that capture a range of use behaviors in our farmer population that represent longer-term and shorter-term phone use and can be measured reliably: total number of contacts saved (i.e., a measure of one's phone-based social network accrued over a longer period of time), number of SMS messages sent and received in the past 24 hours (i.e., a measure of one's phone-based written communication over a very short period of time), and number of calls made and received in the last 24 hours (i.e., a measure of one's phone-based vocal communication over a short period). These measures have been used in other studies of mobile phones in rural East Africa [36, 37]. These 3 variables were square root-transformed in the model due to a high proportion of zero values along with a large number of outlier values. This transformation allowed for adequate scaling and preserved zero-values in the data, though represents a tradeoff as interpretation is less-intuitive than would be with a log-log model [33]. SMS and phone call measures for the last 24 hours served as proxies for previous phone use and were easier for respondents to recall than phone use over a longer period of time. The assumption made in selecting these variables is that phone use over the previous 24 hours is indicative of overall phone use over longer time scales, and that having a greater number of contacts is also indicative of higher levels of phone use.

The phone-for-agriculture model includes a single synthetic focal variable measuring phone use specifically for agricultural purposes to evaluate the question of whether or not the degree of mobile phone use for agricultural activities was associated with self-reported maize yield outcomes (RQ3). Since we were interested in phone use for agricultural purposes generally, and not, for example, using phones to help with fertilizer application specifically, we created a synthetic variable to represent a latent property of our nine individual-level, phone-for-agriculture variables (Table 1). These 9 variables represent different ways that respondents use phones for agriculture-related activities, which we identified through focus group discussions. Unlike the generalized phone use variables (RQ2), the phone-for-agriculture questions were not measured over the past 24 hours, but instead measured binary responses of whether or not respondents use phones for these activities. Collapsing the information in these 9 binary

**Table 1. Regression variable means (and standard errors) stratified by study village.** Binary variables are reported as proportions.

| | Lyamgungwe | | Malagosi | | Kibena | | Mgama | | Total | |
|---|---|---|---|---|---|---|---|---|---|---|
| Dependent variable | | | | | | | | | | |
| Self-reported maize yields in good year (tons/hectare)[1] | 1.81 | (0.17) | 1.70 | (0.20) | 1.37 | (0.11) | 2.12 | (0.26) | 1.73 | (0.10) |
| Independent variables | | | | | | | | | | |
| Phone contacts | 81.57 | (15.46) | 59.16 | (9.91) | 82.52 | (14.49) | 78.39 | (10.76) | 76.30 | (6.62) |
| Total calls in prev. 24hrs | 8.64 | (2.42) | 8.03 | (1.89) | 10.26 | (1.78) | 10.89 | (1.66) | 9.56 | (0.97) |
| Total SMS in prev. 24hrs | 8.90 | (2.69) | 11.11 | (3.05) | 15.01 | (4.25) | 19.24 | (6.30) | 13.81 | (2.23) |
| Phone-for-agriculture composite[2] | 1.24 | (0.14) | 1.25 | (0.11) | 1.18 | (0.11) | 1.46 | (0.08) | 1.28 | (0.06) |
| Wealth score[3] | 5.65 | (0.17) | 5.99 | (0.22) | 6.40 | (0.16) | 6.71 | (0.20) | 6.21 | (0.10) |
| Hectares farmed[1] | 1.12 | (0.09) | 1.48 | (0.18) | 1.03 | (0.12) | 1.32 | (0.14) | 1.22 | (0.07) |
| Proportion male | 0.60 | (0.08) | 0.63 | (0.08) | 0.52 | (0.07) | 0.58 | (0.07) | 0.58 | (0.04) |
| Age | 41.62 | (2.04) | 42.45 | (2.74) | 39.35 | (1.87) | 41.09 | (2.08) | 40.98 | (1.07) |
| Proportion completed primary school | 0.14 | (0.05) | 0.13 | (0.06) | 0.22 | (0.06) | 0.40 | (0.07) | 0.23 | (0.03) |
| n | 42 | | 38 | | 54 | | 45 | | 179 | |

[1] Areas and yields are approximate. Weights and harvested areas were estimated by the respondents.

[2] A synthetic variable representing degree of phone use for agricultural activities, created from HH survey responses by non-linear principal components analysis, and described in the main text.

[3] Wealth score is described in the main text.

variables produced a single continuous variable, trading reduced dimensionality for information loss. Dimension reduction strategies commonly include ordinary principal components analysis or factor analysis [38]. Due to the binary structure of the component variables, we implemented the Gifi method of non-linear principal components analysis [i.e., homogeneity or multiple correspondence analysis; [39]]. This method does not assume normality as does ordinary principal components analysis, yet preserves relative dimensionality of the input matrix. We computed loadings based on the first principal component using the {homals} package in R, producing a single continuous predictor variable, which approximated normal and was not transformed [40].

Both models included an identical set of 5 fixed effect control covariates: a wealth index, farm size, gender, age, and education level. These covariates were informed by literature review and the understood impacts anticipated on agricultural practices and phone use based on various contexts [4, 19, 41]. More generally, these variables describe important dimensions patterning variation in farm-household livelihoods [42]. Wealth is measured through a wealth-poverty index developed for monitoring and evaluation of rural development programs globally, and tailored specifically to Tanzania, derived from 10 survey questions measuring various non-monetary dimensions of wealth and poverty (e.g., female literacy, ownership of durable assets) [43]. The wealth score is scaled to a continuous measure from 0 to 10; raw values are used in the models. Farm size is reported as acres in surveys, or converted; log-transformed values for more appropriate scaling are used in the models. Respondent gender is a binary value for male. Raw values of respondent age are used in the models. Respondent education is a binary value for whether or not they completed primary school.

We implemented likelihood-based multilevel model estimation with the package {lme4} in the R statistical software environment [44]. Models were fitted to data from only those respondents owning their own phone, and for whom complete data existed with respect to variables included in models ($n$ = 179). We evaluated model results by plotting data and coefficient estimates with 95% confidence intervals. Importantly, these not causal models but test only for significant associations. To apportion causality to mobile phones, a randomized controlled trial (RCT) would be ideal.

## Results

Table 2 provides a summary of phone ownership measures stratified by study village. Smartphone ownership and use among respondents was uncommon (6.3% of survey respondents). Thus, the use of communication tools like WhatsApp and Facebook, as well as agricultural information services, was infrequent. During focus group interviews, respondents were asked what types of agricultural activities they conduct with their phones. They reported using phones for everyday activities on the farm including hiring labor or hiring/borrowing equipment, sourcing and buying agricultural inputs, selling agricultural crops, accessing agricultural or weather information, and communication about agriculture. Table 3 presents the percentage of

**Table 2. Phone ownership means (and standard errors) stratified by study village.** Binary variables are reported as proportions.

| Measures | Lyamgungwe | | Malagosi | | Kibena | | Mgama | | Total | |
|---|---|---|---|---|---|---|---|---|---|---|
| Phone within the household (%) | 88.57 | (3.83) | 94.52 | (2.68) | 97.14 | (2.01) | 98.48 | (1.52) | 94.62 | (1.35) |
| Own a phone (%) | 77.14 | (5.06) | 57.53 | (5.83) | 88.57 | (3.83) | 78.79 | (5.07) | 75.27 | (2.59) |
| Own an internet capable phone (%) | 17.14 | (4.54) | 16.44 | (4.37) | 35.71 | (5.77) | 18.18 | (4.78) | 21.86 | (2.48) |
| Own a smartphone (%) | 2.86 | (2.01) | 1.37 | (1.37) | 8.57 | (3.37) | 7.58 | (3.28) | 5.02 | (1.31) |
| n | 70 | | 73 | | 70 | | 66 | | 279 | |

Table 3. Percentages of respondents using their own mobile phones for specific agricultural purposes.

| Purposes | % of respondents (N = 179) |
| --- | --- |
| Discussions with friends and relatives about agriculture | 75 |
| Selling crops | 70 |
| Talking to agricultural extension agent | 65 |
| Buying seeds or fertilizer | 62 |
| Gathering information about agricultural practices | 65 |
| Hiring or borrowing equipment | 48 |
| Using mobile money services | 47 |
| Accessing weather information | 48 |
| Hiring farm labor | 50 |

farmers in each of the 4 study villages who use phones in each of these ways. Importantly, 20% of respondents reported using their phones for all the agricultural purposes of interest, while 25% reported using their phone for none of these purposes.

Additionally, addressing RQ1, respondents were asked how mobile phones were affecting their own agricultural productivity. Respondents were free to respond based on their own experiences and interpretations of the questions. Approximately 47% of respondents stated that the use of a phone has reduced the amount of time they spent buying inputs or selling crops, and 50% of respondents reported that the use of a phone has reduced the amount of money they spent on farm activities. Further, 64% reported that the use of a phone has increased profits from farming compared to when the respondent did not have a phone. The percentage of respondents who answered yes to each of these questions indicates that for many, phones have increased the efficiency and cost-effectiveness of farming by reducing the time and money spent on farming activities, while simultaneously increasing profits.

Addressing RQ2, model results suggest general phone use has inconsistent associations with self-reported maize yield (Fig 2; Tables 1 and 4). The number of farmers' phone contacts, the number of recent calls, and the number of recent SMS messages do not have credible associations with reported yield with a 95% CI. Farmers in Kibena Village have the lowest self-reported yields, and Lyamgungwe the highest, though these differences are relatively small,

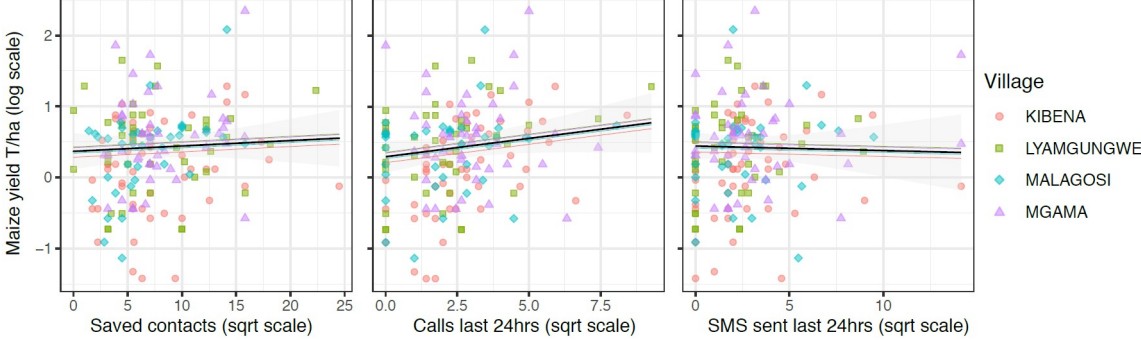

Fig 2. General phone use model estimates for the predicted association between reported maize yield outcomes and 3 phone use predictors (RQ2). Coefficient estimates of phone contacts (left), calls in the last 24 hours (center), and SMS sent in the last 24 hours (right) have mixed associations with the self-reported maize yield outcome, after controlling for individual-level wealth, gender, farm size, age, and education. The model is estimated with varying intercept effects at the household and village levels; village intercepts are plotted as colored lines in colors corresponding to village data points. Coefficients are plotted separately as predictions with 95% confidence intervals, with other fixed effect variables held at mean or modal values, and with averaging over uncertainty estimated across the model. See Table 4 for model estimates.

**Table 4. Model estimates and [95% confidence intervals[1]] for Figs 2 and 3.**

| | Maize yield (Fig 2; RQ2) | | Maize yield (Fig 3; RQ3) | |
|---|---|---|---|---|
| Grand intercept | -0.070 | [-0.608, 0.467] | -0.467 | [-0.988, 0.055] |
| Fixed effects | | | | |
| # of contacts | 0.008 | [-0.016, 0.031] | | |
| # of calls[2] | 0.052 | [-0.008, 0.111] | | |
| # of SMS[2] | -0.006 | [-0.050, 0.038] | | |
| Synthetic phone use variable | | | 0.273 | [0.162, 0.383] |
| Wealth score | 0.035 | [-0.035, 0.106] | 0.049 | [-0.016, 0.114] |
| Hectares farmed | 0.339 | [0.186, 0.492] | 0.325 | [0.182, 0.468] |
| Gender (male) | 0.096 | [-0.093, 0.285] | 0.141 | [-0.018, 0.301] |
| Age | 0.000 | [-0.007, 0.007] | 0.002 | [-0.004, 0.008] |
| Education | -0.074 | [-0.288, 0.139] | -0.016 | [-0.218, 0.185] |
| Variance on varying intercept effects | | | | |
| Household | 0.085 | | 0.035 | |
| Village | 0.009 | | 0.005 | |
| Observations | 179 | | 179 | |
| Model marginal $R^2$ | 0.195 | | 0.284 | |
| Model conditional $R^2$ | 0.425 | | 0.382 | |

[1] Confidence intervals (CIs) here indicate 95% confidence that the mean of the variable for all respondents lies within the reported interval. Estimates that are significant at this level have CIs that do not cross 0.

[2] Made/sent and received in the past 24 hours

which is apparent in the varying effects estimates from the model-averaged predictions plots (Fig 2, colored lines).

Addressing RQ3, when we examine phone use specific to agricultural activities, the results suggest that phone use for agriculture is credibly associated with higher self-reported maize yield (Fig 3; Tables 1 and 4). This result is consistent across the sample after controlling for observed differences among individuals in wealth, farm size, gender, age, and education, and also after controlling for unobserved differences within households and within villages. Kibena is estimated to have the lowest yields, and Lyamgungwe the highest, as shown by the village-level varying intercept adjustments.

## Discussion

Our goal was to examine relationships between mobile phone use and agricultural productivity at the household and farmer levels. A key result is the positive association between phone use for agricultural activities and self-reported agricultural yields (RQ3). Further, our results find that many farmers had positive perceptions of mobile phone use increasing agricultural efficiency through increasing profits, decreasing costs, and decreasing time investments in farming (RQ1). However, our findings showed no consistent associations between general phone use and self-reported maize yield, when phone use is measured as number of contacts, number of SMS sent and received, and number of phone calls made and received within a narrow window of time (RQ2).

Our finding, that the association between yields and general phone, is not statistically significant is not necessarily surprising given that much of the other research on small-holder agricultural outcomes has focused on the use of phones to convey market and weather information [21, 22]. Alternatively, some confounding variable, such as social network, could drive

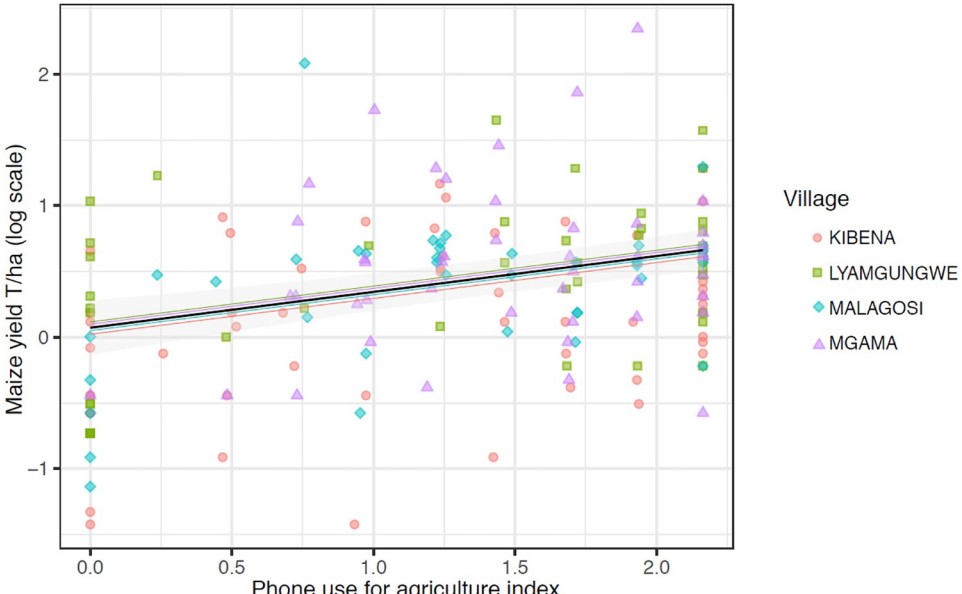

**Fig 3. Phone for agriculture model estimates for the predicted association between reported maize yield outcomes and the degree to which farmers integrate phones into agricultural activities (RQ3).** A synthetic variable representing degree of phone use for agricultural activities was created from household survey responses by non-linear principal components analysis. Phone use for agriculture has a positive association with self-reported maize yield, after controlling for individual-level wealth, gender, farm size, age, and education. The model is estimated with varying intercept effects at the household and village levels; village intercepts are plotted as colored lines in colors corresponding to village data points. The phone use coefficient is plotted with 95% confidence intervals, with other fixed effect variables held at mean or modal values, and with averaging over uncertainty estimated across the model. See Table 4 for model estimates.

both phone use and agricultural productivity, though in this context respondents did not highlight this issue during focus group interviews. Notably, qualitative analyses of phone use in East Africa have identified that daily phone-use is commonly, simply to connect with friends and family [32, 37]. A lesson here may be that given the breadth of ways that phones are used, general phone use is a poor predictor of specific economic outcomes. Also, this finding speaks to the difficulty of measuring general phone use over extended time-periods (as we discuss in the Limitations section).

Still, this result suggests that simply owning and using a mobile phone may not be enough to support agricultural productivity. Instead, how a farmer uses her phone may be critical. Other research has identified potential mechanisms behind positive relationships between using phone use and for agricultural productivity, which include: the use of mobile phones for connecting farmers to buyers [6], acquiring inputs for farming [45], reducing transaction costs and time associated with agricultural activities [46, 47], and exchanging agricultural information and recommendations [4, 48]. Our results are consistent with these findings, specifically our observations of respondents' positive perceptions of mobile phones for decreasing time and money spent, and increasing profits from agricultural activities. Along these lines, a farmer can use his phone to communicate with a fertilizer seller in town, buy fertilizer, and then recruit a friend to help him transport the fertilizer to the farm, saving both time and money.

Our findings are also consistent with those of studies connecting mobile phone use and increased agricultural yield. A study in India [49] found that 35% of farmers who used their phones for connecting with markets, getting better prices, and getting agricultural information

reported increased yields. A study of coffee farms in Uganda found positive associations between mobile phone use and increased coffee harvests, as well as higher off-farm incomes [19]. And a study in Ghana [20] concluded that a farmer with a mobile phone had, on average, an increased maize yield of 261 kg/ha per production season compared to farmers without a phone. However, these studies all employed different metrics in measuring phone use, which can have an important impact on the results, as our study shows. Ultimately, to better understand the causal mechanisms in our study area would require more involved methodological approaches, including in-depth ethnographic work, and/or an RCT.

Overall, our findings support the targeted and intentional use of ICTs as a strategy to improve agricultural productivity and economic development. Mobile phones can support development by increasing household management efficiency [1, 16, 50] and contributing to existing livelihood activities [37]. They can provide people and communities in rural parts of the developing world with access to digital information and resources, as well as new types of knowledge sharing platforms [51, 52]. Our results highlight the importance of mobile phones at individual- and household-scales, and complement research at larger scales, which highlights how mobile phone penetration and use can improve agricultural market performance in developing economies [16].

## Policy implications

The results presented in this paper highlight 2 important areas for potential future policy interventions aimed at improving agricultural productivity for smallholder farmers while providing other social and economic benefits [53]. First, simply owning and using a mobile phone may have little impact on improving smallholders' yields. Alternatively, our findings suggest that when farmers' use phones specifically and intentionally for a range of agricultural tasks, yields can improve. Consequently, governmental and non-governmental interventions should encourage smallholders to use phones *specifically* for tasks throughout the agricultural enterprise to increase the likelihood that phone use has a positive impact on yield. This could include strategies that promote farmer education on the uses of ICTs for agriculture, as well as the development of extension services via mobile phones. For example, a national governmental agricultural extension agency could promote technology trainings to farmers, as well as extension services via mobile phone technologies, such as SMS and call-in services. These types of policies may also enable the extension agency to reach out to a greater number of farmers, and provide easy access to important agricultural information [13, 15].

Second, the results highlight the importance of perception in the adoption of new technologies that may be promoted by various policies. Often, regardless of what policy interventions are implemented, critical to the adoption of any innovation are the perceived benefits from stakeholders [26]. The results presented in this paper show that many respondents had positive perceptions of mobile phone use for agriculture, which suggests that they are ready to adopt greater mobile phone use within the agricultural sector. Thus, timely policy interventions would likely be well received within the study communities. This also highlights that in other contexts, studies of stakeholder perceptions of any intervention are important to understand the likely success of a given policy.

## Limitations and future directions

First, in controlling for demographic characteristics including farm size, wealth, and gender, our goal was to gain a basic understanding of the relationship between phone use expressly for agricultural purposes and self-reported yield, but not to examine other associations. Also, we did not control for the influence of other ICTs such as radio or television, largely due to the

limited access to electricity for most respondents. Furthermore, we did not include more nuanced control variables such as 'entrepreneurship' and 'innovativeness', which are culturally relative and difficult to measure. However, the topic of differential impacts of ICTs is something we plan to explore in future work, especially given the growing body of research focused on the 'digital divide' in access and use of ICTs between men and women [4, 24].

Second, measuring both the character and volume of phone use over long periods is also challenging. Research respondents' abilities to recall phone use over long periods is low and soliciting this information can lead to estimation errors. In this study, we avoided this by getting reliable measures over a narrow amount of time. Also, measuring phone use specifically for agricultural activities can be complicated by the seasonality of the agricultural cycle. Accordingly, future research on phone use should examine the temporal nature of agricultural practices in order to more effectively measure the impact of phone use on agricultural productivity throughout the agricultural cycle. For this study, data collection took place at the end of the harvest season.

Third, while many respondents reported greater profits from agriculture and less money spent on agricultural activities through the use of a mobile phone, this does not necessarily factor in the costs of mobile phone ownership and use itself. Owning and using a mobile phone does have financial costs which can include purchasing of the phone, buying phone credit, and paying for phone charging services. These costs were not accounted for in this study, and could possibly influence the costs and benefits of using mobile phones for agricultural activities if the cost of owning and using a mobile phone exceeds the financial benefits from using phones for agricultural activities.

Lastly, it is important to note that while phone use is mainly an individual activity, agricultural productivity is generally a household outcome. Therefore, there is a mismatch in scale between phone use and agricultural productivity. In this study we aimed to address this by interviewing both male and female household members. However, innovative methods for studying phenomena at these 2 different scales may help better address these issues in the future.

## Conclusions

Addressing the objectives of this paper, we conclude the following: (RQ1) many farmers had positive perceptions about the benefits of mobile phones for their agricultural productivity; (RQ2) there was not a significant relationship between general mobile phone use and self-reported maize yield; and (RQ3) there was a positive significant relationship between mobile phone use for agricultural activities and self-reported maize yield.

Our research indicates that there are significant policy opportunities to leverage the existing use of ICTs to increase efficiency, yields, and profits, by better directing the use of mobile phones towards agricultural activities. This potential will grow as phone use continues to expand and new agricultural strategies and technologies are developed. However, technology-based policy interventions are not panaceas and need to be part of comprehensive strategies for rural economic development including investments in physical infrastructure, education, health services, and access to electricity [54].

## Acknowledgments

We are grateful to smallholder farmers who took time out of their days to contribute participate in this research. We are also grateful to our survey enumerators who spent 3 weeks in the field conducting surveys in the 4 communities.

## Author Contributions

**Conceptualization:** Jonathan D. Salerno, Jason C. Neff, Joel Hartter.

**Data curation:** Jonathan D. Salerno.

**Formal analysis:** Amy Quandt, Jonathan D. Salerno, Emilie Xu.

**Funding acquisition:** Jonathan D. Salerno, Joel Hartter.

**Investigation:** Amy Quandt, J. Terrence McCabe, Joel Hartter.

**Methodology:** Amy Quandt, Jason C. Neff, Timothy D. Baird, Jeffrey E. Herrick, J. Terrence McCabe, Joel Hartter.

**Project administration:** Amy Quandt, Jonathan D. Salerno, Joel Hartter.

**Resources:** Amy Quandt, Joel Hartter.

**Supervision:** Amy Quandt, Jason C. Neff, Joel Hartter.

**Validation:** Amy Quandt, Jonathan D. Salerno, Emilie Xu.

**Visualization:** Jonathan D. Salerno, Timothy D. Baird.

**Writing – original draft:** Amy Quandt, Jonathan D. Salerno, Jason C. Neff, Jeffrey E. Herrick, J. Terrence McCabe, Joel Hartter.

**Writing – review & editing:** Amy Quandt, Jonathan D. Salerno, Jason C. Neff, Timothy D. Baird, Jeffrey E. Herrick, J. Terrence McCabe, Joel Hartter.

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
