## [Decision Letter · Decision Letter 0]

28 Apr 2020

PONE-D-20-05740

Mobile phone use associated with higher smallholder agricultural productivity in Tanzania, East Africa

PLOS ONE

Dear Dr. Quandt,

I hope you are well in these challenging times. I have now received review reports from two experts in the field. Based on these and my own reading, I have decided to request some revisions to be made. You will see that one of the reviewer starts out pretty critical, noting that still a lot of work will be needed to make the manuscript suitable for publication. However, you will also see that many of this reviewers comments are not all that substantial, mostly related to the presentation. I would therefore suggest to focus more on the comments of the other reviewer, that is more positive but points out a few issues that needs to be address.

For my own reading, I like the random effects model, but also wonder if a simple linear (pooled) regression, or a regression with fixed effects for the villages would yield conclusions that differ much. If the latter is the case, reasons should be explored.

I agree with the reviewer that variables should not be excluded just because they are not significant; you should be guided by theory.

While you are mostly careful not to attribute a causal interpretation to the relationships you find, I think it would not hurt to have an extra paragraph making explicit that, ideally, these sort of questions should be investigated using an RCT or other method that can isolate exogenous variation in phone use. Related, I agree with the reviewer that you can not control for (un)observables (something you later also admit) by including wealth in your regression: wealth is endogenous.

Finally, you combine different dimensions of phone use into one indicator. But number of contacts in phone may measure a very different attribute than eg calls placed. The first one may be a proxy for social network, which has also been found to yield increments for many reasons (access to finance,...). Thus, finding a correlation may not be because of phone use but due to social network effects. This will have implications for policy.

I am thus more optimistic than reviewer 2 and think it should not take too much effort to respond to these issues. I do want to stress though, that I will expect to see some additional regressions in the response to reviewers such that we get a sense of the robustness of the findings, that is, that results are not driven by the small sample.

To enhance the reproducibility of your results, we recommend that if applicable you deposit your laboratory protocols in protocols.io, where a protocol can be assigned its own identifier (DOI) such that it can be cited independently in the future. For instructions see: http://journals.plos.org/plosone/s/submission-guidelines#loc-laboratory-protocols

We look forward to receiving your revised manuscript.

Kind regards,

Bjorn Van Campenhout, Ph.D.

Academic Editor

PLOS ONE

Journal Requirements:

Reviewers' comments:

Reviewer's Responses to Questions

**Comments to the Author**

1. Is the manuscript technically sound, and do the data support the conclusions?

Reviewer #1: Partly

Reviewer #2: Partly

2. Has the statistical analysis been performed appropriately and rigorously? 

Reviewer #1: No

Reviewer #2: No

3. Have the authors made all data underlying the findings in their manuscript fully available?

Reviewer #1: Yes

Reviewer #2: No

4. Is the manuscript presented in an intelligible fashion and written in standard English?

Reviewer #1: Yes

Reviewer #2: No

5. Review Comments to the Author

Reviewer #1: Review comments

Manuscript ID – PONE – D – 20 – 05740: Title: Mobile Phone use Associated with Higher Smallholder Agricultural Productivity in Tanzania – East Africa. Due date: 3rd April 2020

General Comments:

Good focus of the paper and with a clear new novelty (Perceptions on MP use versus agricultural productivity. Well written and referenced but a few technical comments highlighted below if addressed would make the paper better. Some robustness checks would be so nice to include (a different method of analysis to back up similar results as these results here)

Specific Comments:

TITLE:

1. I think there should be a “,” and “is” between “use” and “associated”

ABSTRACT

2. Generally comprehensive and easy to understand

3. there should be some highlights on the data sample size used,

4. and methods used to analyze the data should also be concisely highlighted in the abstract

INTRODUCTION

1. Generally, well written, referenced and easy to understand.

2. However – villages of interest in Tanzania are not mentioned at all, or

3. Perhaps some specific statistics of this Iringa province, since all statistics mentioned here are only at regional or government level. Provincial or village specific stats, would make the write up stronger.

4. Novelty is clear that, the paper focuses on farmers’ perceptions on agricultural productivity, which has rarely been researched

5. A good mix of qualitative and quantitative approach

METHODS

1. Some specific statistics about the four villages – would be interesting to look at here briefly

2. Line 178 – what is meant by “household diversity?” Technically this is largely used with regards to household production diversity or household dietary diversity, so it may be confusing here – and you may need another term to avert the potential confusion to readers

3. Line 187 – what informed the sample size of 279? Some background could be helpful

4. Line 204 – how was it likely to influence the responses by having same gender enumerator/respondents? This needs to be explained as it is not usual, and would be a potential source of biases

5. Lines 288 – 290 – variables age and education based on several literatures can influence mobile phone use, as well as agricultural yield. Moreover, they also make logical sense to be controlled for other than being merely excluded because they were not significant. This would deny us potentially logical and economic sense/significance based modelling which is another excellent component for validity of results on top of statistical significance. I would wish to see how the model results come out with these logical, and economically valid variables included

6. Also, a wealth index and use of mobile phones, could be endogenous; how was this potential endogeneity handled?

7. Line 295 – use either 1 to 100 or one to a hundred – to avert potential confusion

8. Line 296 – was farm-size transformed because it was not normal? If yes, this has to be stated, but not only stated for a few variables

9. Lines 309 – 311 – the assumption of number of calls or SMSs reflecting long time use of the phone; did you control for the seasonality aspect? I would assume that during the planting or harvesting seasons, phone calls would be more. Perhaps, you may need to clarify on the seasonality aspect with regards to this assumption

RESULTS

1. Table 1; were these statistics different across the 4 villages? It would be nice to show these. Also, in discussions of these results, it would be good to briefly show how these compare with national statistics or regional or provincial ones.

2. In Table 2, presenting results in percentages would make it easier for comparison and understanding

3. Table 3 – for ease of understanding results in line with usual literature – could you show the significance levels of each of the control variables by asterisk? Also reduce the grid lines in the table for more neatness.

4. Age and Education should also be controlled for in the results of this table 3 for their potentially significant logical and economic importance.

DISCUSSIONS

1. Line 404 – could you be specific with these confounding factors that contrary to general literature – led the results of general MP use not to be associated significantly with yield? This would make the paper more “self-contained” and independent

2. In Policy Implications – perhaps you must strongly make it clear that this advise is intended to “smallholder farmers” who are investigated in this paper. Otherwise – blindly advising policy to generally support MP use on specific agricultural activities may not have similar results for cattle herdsmen who are also common in rural Tanzania.

CONCLUSIONS

1. There was only about 4% of the sample using smart phones – and why make this a priority concluding remark in line 528 – in the first real paragraph that should be aimed at the central results of the study.

Reviewer #2: The attachment provides a detailed analysis of my my thinking about this manuscript. The reviewer thinks that with a very major revision, the paper can improve and substantially be at the point where it can be acceptable for publication. Authors would have to do a lot work, though in in this period, in order to get the manuscript to an acceptable level.

I encourage them to follow the points raised in my reviewer comments very carefully. and I will be willing and hopefully, available to to do another round of review of this paper if needed.

6. PLOS authors have the option to publish the peer review history of their article (what does this mean?). If published, this will include your full peer review and any attached files.

Reviewer #1: Yes: Dr. Haruna Sekabira

Reviewer #2: No

---

## [Author Response · Author response to Decision Letter 0]

9 Jul 2020

RESPONSE TO REVIEWERS

Editor comments

I hope you are well in these challenging times. I have now received review reports from two experts in the field. Based on these and my own reading, I have decided to request some revisions to be made. You will see that one of the reviewer starts out pretty critical, noting that still a lot of work will be needed to make the manuscript suitable for publication. However, you will also see that many of this reviewers comments are not all that substantial, mostly related to the presentation. I would therefore suggest to focus more on the comments of the other reviewer, that is more positive but points out a few issues that needs to be address.

RESPONSE: Thank you for your effort to review our paper. We appreciate the thoughtful comments and have worked hard to incorporate them into our revised paper. Along these lines, we have made substantive revisions to every section of the paper to improve the quality of the writing and the clarity of our communication. We have made revisions to the modeling and presentation of data and findings, including overhauls of each of the tables and figures. We believe these revisions, which are described in greater detail throughout our response to reviewers’ comments, greatly improve the quality of the paper.

EDITOR: For my own reading, I like the random effects model, but also wonder if a simple linear (pooled) regression, or a regression with fixed effects for the villages would yield conclusions that differ much. If the latter is the case, reasons should be explored.

RESPONSE: We estimated 2 alternatives to each of the models presented in the revision, the first a simple linear model without the random effects for village (leaving out village effects entirely), and the second a regression with fixed effects for village. Estimates for focal variables and controls were relatively the same across revised presented models and these alternatives (i.e., coefficient estimates and standard errors differed slightly, but credibility or significance of estimates were the same). That is, for our general phone use model only farmed area was credibly associated with reported maize yield; and, for our phone use for agriculture model, phone-for-agriculture and farmed area were credibly associated with reported maize yield. Therefore, we do not report these additional models and rely on our a priori justification for using a multilevel model with random effects for village. We do, however, note in the Methods that our reported model specification ranks higher by AICc information criterion than the alternative, and include dAICc scores. However, if the editor feels strongly that the alternative models should be included in the revised text, then we are happy to comply.

EDITOR: I agree with the reviewer that variables should not be excluded just because they are not significant; you should be guided by theory.

RESPONSE: We agree that age and education level serve as good additional controls, and we have included them in the revised models. Notably, they were not excluded from the original submission based on the estimates, rather we opted for a more parsimonious model. The original text was not clear on this point. However, we agree that these additional variables are wise to include in both revised models. Estimates of our focal variables remain relatively unchanged.

EDITOR: While you are mostly careful not to attribute a causal interpretation to the relationships you find, I think it would not hurt to have an extra paragraph making explicit that, ideally, these sort of questions should be investigated using an RCT or other method that can isolate exogenous variation in phone use. Related, I agree with the reviewer that you can not control for (un)observables (something you later also admit) by including wealth in your regression: wealth is endogenous.

RESPONSE: While economists are often concerned about exogeneity, other fields have different perspectives. Our group, which is composed primarily of human geographers and anthropologists, derives some confidence in our study from our qualitative focus groups conducted in the field to understand mechanisms and ultimately shape the design of our survey. Still, as noted, we avoid “causal” language opting instead for language of “relationships” or “associations.” Also, we now include a clear statement at the end of the methods section that describes how the models are not causal, and that an RCT would be ideal. 

EDITOR: Finally, you combine different dimensions of phone use into one indicator. But number of contacts in phone may measure a very different attribute than eg calls placed. The first one may be a proxy for social network, which has also been found to yield increments for many reasons (access to finance,...). Thus, finding a correlation may not be because of phone use but due to social network effects. This will have implications for policy.

RESPONSE: The original text was not entirely clear. We have revised the model description in the Methods section, and restructured the Results for better clarity. Contacts, calls, and texts are estimated separately in the general phone use model (see Fig. 3). Also, we’ve discussed these in greater detail in the methods. And we’ve discussed the potential for social networks to serve as an omitted causal variable in the discussion section. In the phone for agriculture model, we do combine multiple measures of agricultural phone use into an index measure as the focal variable, including using phones for accessing information on fertilizer or hiring farm labor. Lastly, we’ve revised the policy implications section to be more specific and conservative.

EDITOR: I am thus more optimistic than reviewer 2 and think it should not take too much effort to respond to these issues. I do want to stress though, that I will expect to see some additional regressions in the response to reviewers such that we get a sense of the robustness of the findings, that is, that results are not driven by the small sample.

RESPONSE: We note the additional models estimated and revisions made in our response just above. Here we include fixed effect coef estimates from each reported models and the two alternatives to each.

General phone use model, as reported in revision, with village random/varying effects:

# Estimate Std. Error t value

#(Intercept) -0.0703427 0.2741156 -0.257

#sqrtPhContacts 0.0075574 0.0118448 0.638

#sqrtCallTotal 0.0515576 0.0304081 1.696

#sqrtSmsTotal -0.0063911 0.0224546 -0.285

#scaledPovertyScore 0.0351852 0.0358823 0.981

#logFarmHectPlant 0.3388541 0.0781271 4.337*

#male 0.0963103 0.0963723 0.999

#as.numeric(age) 0.0001625 0.0034560 0.047

#edLevSec -0.0741210 0.1089327 -0.680

General phone use model, with no village effects

# Estimate Std. Error t value

#(Intercept) -0.0449657 0.2669384 -0.168

#sqrtPhContacts 0.0081260 0.0118962 0.683

#sqrtCallTotal 0.0498101 0.0305475 1.631

#sqrtSmsTotal -0.0073511 0.0226110 -0.325

#scaledPovertyScore 0.0292653 0.0355737 0.823

#logFarmHectPlant 0.3553133 0.0768808 4.622*

#male 0.0992511 0.0965557 1.028

#age 0.0002841 0.0034790 0.082

#edLevSec -0.0603360 0.1087211 -0.555

General phone use model, with village fixed effects:

# Estimate Std. Error t value

#(Intercept) -0.2462413 0.2977124 -0.827

#sqrtPhContacts 0.0070669 0.0118938 0.594

#sqrtCallTotal 0.0527244 0.0305543 1.726

#sqrtSmsTotal -0.0051489 0.0225070 -0.229

#scaledPovertyScore 0.0398582 0.0366041 1.089

#logFarmHectPlant 0.3260410 0.0800655 4.072*

#male 0.0930076 0.0969894 0.959

#age 0.0001362 0.0034648 0.039

#edLevSec -0.0844512 0.1100695 -0.767

#as.factor(village)LYAMGUNGWE 0.2711080 0.1308731 2.072*

#as.factor(village)MALAGOSI 0.0986482 0.1363669 0.723

#as.factor(village)MGAMA 0.2440906 0.1261938 1.934*

Phone for agriculture model, as reported in revision, with village random/varying effects:

# Estimate Std. Error t value

#(Intercept) -0.466821 0.266043 -1.755

#phAgLoad 0.272561 0.056312 4.840*

#scaledPovertyScore 0.048771 0.033043 1.476

#logFarmHectPlant 0.325143 0.073103 4.448*

#male1 0.141193 0.081337 1.736

#as.numeric(age) 0.002237 0.003139 0.712

#edLevSec -0.016479 0.102698 -0.160

Phone for agriculture model, with no village effects

# Estimate Std. Error t value

#(Intercept) -0.456685 0.261457 -1.747

#phAgLoad 0.276297 0.056434 4.896*

#scaledPovertyScore 0.044510 0.032651 1.363

#logFarmHectPlant 0.334173 0.072170 4.630*

#male1 0.141747 0.081587 1.737

#as.numeric(age) 0.002407 0.003152 0.764

#edLevSec -0.008811 0.102058 -0.086

Phone for agriculture model, with village fixed effects:

# Estimate Std. Error t value

#(Intercept) -0.593713 0.285615 -2.079

#phAgLoad 0.265868 0.056733 4.686*

#scaledPovertyScore 0.054234 0.034055 1.593

#logFarmHectPlant 0.314582 0.075212 4.183*

#male1 0.140365 0.081525 1.722

#as.numeric(age) 0.002033 0.003149 0.646

#edLevSec -0.025678 0.104654 -0.245

#as.factor(village)LYAMGUNGWE 0.225751 0.118982 1.897(*)

#as.factor(village)MALAGOSI 0.058243 0.123694 0.471

#as.factor(village)MGAMA 0.179576 0.116765 1.538

Regarding additional changes not noted below, we recalculated and reformatted tables. Due to a missing value, recalculated means and standard errors for certain variables are slightly different from those reported in the original submission. Missing values are dropped by default when estimating regression models with lme4 in R, and so regression estimates were unaffected.

Reviewer 1 comments

General Comments:

Good focus of the paper and with a clear new novelty (Perceptions on MP use versus agricultural productivity. Well written and referenced but a few technical comments highlighted below if addressed would make the paper better.

Specific Comments:

Title

1. I think there should be a “,” and “is” between “use” and “associated”

RESPONSE: We have added “is” but don’t see how the “,” makes sense grammatically.

Abstract

2. Generally comprehensive and easy to understand 

3. there should be some highlights on the data sample size used, 

4. and methods used to analyze the data should also be concisely highlighted in the abstract

RESPONSE: We agree, and implement these changes in the revised abstract.

Introduction

1. Generally well written, referenced and easy to understand.

2. However – villages of interest in Tanzania are not mentioned at all, or 

3. Perhaps some specific statistics of this Iringa province, since all statistics mentioned here are only at regional or government level. Provincial or village specific stats, would make the write up stronger.

4. Novelty is clear that, the paper focuses on farmers’ perceptions on agricultural productivity, which has rarely been researched

5. A good mix of qualitative and quantitative approach

RESPONSE: We’ve provided Iringa specific statistics in the Introduction. 

Methods

1. Some specific statistics about the four villages – would be interesting to look at here briefly 

RESPONSE: We’ve included district-level statistics (agriculture, education, phone ownership) in the study-site section of the methods. 

2. Line 178 – what is meant by “household diversity?” Technically this is largely used with regards to household production diversity or household dietary diversity, so it may be confusing here – and you may need another term to avert the potential confusion to readers

RESPONSE: Indeed, we refer here to intra-household, economic diversity. This language, along with some surrounding language, has been changed to be clearer. 

3. Line 187 – what informed the sample size of 279? Some background could be helpful

RESPONSE: Revised text details a goal sample of 40 household compounds per village, which was informed by previous research implementing similar analytical methods; multiple individuals were surveys in each household.

4. Line 204 – how was it likely to influence the responses by having same gender enumerator/respondents? This needs to be explained as it is not usual, and would be a potential source of biases

RESPONSE: Conversely, our experience working in this area has taught us that having male enumerators for female respondents can be problematic due to cultural norms. We now include the following language in the text: “Using same gender enumerators is a common practice in rural, developing areas where education is relatively low and gender roles can be hierarchical.”

5. Lines 288 – 290 – variables age and education based on several literatures can influence mobile phone use, as well as agricultural yield. Moreover, they also make logical sense to be controlled for other than being merely excluded because they were not significant. This would deny us potentially logical and economic sense/significance based modelling which is another excellent component for validity of results on top of statistical significance. I would wish to see how the model results come out with these logical, and economically valid variables included

RESPONSE: We agree that age and education level would serve as good additional controls, and we have included them in the revised models. Notably, they were not excluded from the original submission based on the estimates, rather we opted for a more parsimonious model. The original text was incorrectly stated. However, we agree that these additional variables are wise to include in both revised models. Estimates of our focal variables remain relatively unchanged.

6. Also, a wealth index and use of mobile phones, could be endogenous; how was this potential endogeneity handled?

RESPONSE: Through our qualitative focus group interviews, we learned that wealth/poverty are important drivers of yield, as they affect smallholders’ ability to acquire farm inputs. We also learned that phones help farmers to be in touch with extension agents who can provide useful information. These responses give us confidence that the wealth and phone-use represent distinct mechanisms that are directly related to yield. Also, we are clear throughout the paper that we are not presenting causal models. Still, our models include covariates that meaningfully describe maize yields in these small-holder farming systems (i.e., wealth, acres, location, etc.). With our analyses, we’re looking to explain additional variance in yield that can be apportioned to variance in phone use. Finally, the wealth index used is a multi-dimensional measure of wealth and poverty, which we now articulate in the revised text rather than only supplying the reference. 

7. Line 295 – use either 1 to 100 or one to a hundred – to avert potential confusion 

RESPONSE: We’ve changed this to “1 to 100”

8. Line 296 – was farm-size transformed because it was not normal? If yes, this has to be stated, but not only stated for a few variables

RESPONSE: Substantial revisions to the Methods make each variable transformation clear.

9. Lines 309 – 311 – the assumption of number of calls or SMSs reflecting long time use of the phone; did you control for the seasonality aspect? I would assume that during the planting or harvesting seasons, phone calls would be more. Perhaps, you may need to clarify on the seasonality aspect with regards to this assumption

RESPONSE: We don’t assume that SMS reflect phone use over a long period of time. Instead, it serves as a proxy of phone use during a short period of time, as we now indicate in text. Similarly, phone calls also serve as a proxy for use (vocal rather than written communication) over a short period. We use phone-contact list as a proxy of use over a comparatively longer period. We are now clear about this in the text. We also now acknowledge the issue of seasonality. And notably, our measurements are taken during an agricultural active season - harvest season. 

Results

1. Table 1; were these statistics different across the 4 villages? It would be nice to show these. Also, in discussions of these results, it would be good to briefly show how these compare with national statistics or regional or provincial ones.

RESPONSE: We understand this comment to mean that the reviewer is looking for significance tests on means across villages. This information is not central to our research questions and reporting these statistics would overly complicate the table, and burden the reader. We do however report village-level means with standard errors to give interested readers a sense of how data are patterned across villages. 

2. In Table 2, presenting results in percentages would make it easier for comparison and understanding

RESPONSE: Table 2 does report whole number percentages. We’ve made a small adjustment to the column heading so that it’s easier to read.

3. Table 3 – for ease of understanding results in line with usual literature – could you show the significance levels of each of the control variables by asterisk? Also reduce the grid lines in the table for more neatness. 

RESPONSE: We avoid categorizing coefficient estimates as “significant” in order to avoid confusion with null hypothesis significance testing and frequentist approaches. Rather, models make estimates with associated standard errors and confidence intervals, which we report at the 95% level. We opt for this approach following shifting norms of practice away from statistical significance in line with the American Statistical Association and numerous publishing groups . To help with understanding and readability, we bold coefficients estimated as credible at 95% confidence in the Table. Likewise, we reformatted tables for easier reading in review.

4. Age and Education should also be controlled for in the results of this table 3 for their potentially significant logical and economic importance. 

RESPONSE: As stated in the response above, age and education are included as additional controls in the revised models.

Discussions

1. Line 404 – could you be specific with these confounding factors that contrary to general literature – led the results of general MP use not to be associated significantly with yield? This would make the paper more “self-contained” and independent 

RESPONSE: We’ve changed this section, replacing 3-4 sentences with new language that better addresses how our findings fit with existing research. We explain how the non-significance of our general phone-use variable is not surprising when compared with other studies, which have focused on market and weather information. And we cite other research that suggests that most phone use is not about specific economic activities. We feel that this is a much improved interpretation of this non-significance.

2. In Policy Implications – perhaps you must strongly make it clear that this advise is intended to “smallholder farmers” who are investigated in this paper. Otherwise – blindly advising policy to generally support MP use on specific agricultural activities may not have similar results for cattle herdsmen who are also common in rural Tanzania.

RESPONSE: We have taken this advice and included specific language about small-holder farmers throughout the section. Also, we are more specific in our advice, noting that intentional phone use across a range of agricultural tasks is most likely to positively impact yields. 

Conclusions

1. There was only about 4% of the sample using smart phones – and why make this a priority concluding remark in line 528 – in the first real paragraph that should be aimed at the central results of the study.

RESPONSE: We agree, and have shifted this concluding paragraph back to a focus on mobile phones broadly. 

Reviewer 2 comments 

This article seeks to “examine relationships between mobile phone use and agricultural productivity at the household and farmer levels” in Tanzania. Authors utilize multi-level modeling and principal component analysis to examine the nature and type of relationships between farmers’ perception and mobile phone use in agriculture, and the level of agricultural productivity, measured by maize yields per hectare. There are three findings: First, a positive relationship between mobile utilization and agricultural yields. Second, a positive relationship between farmer-perception of mobile phones and “agricultural efficiency” in terms of lower costs, higher profits, and lower “time investments in farming”. The third finding is that there is “no-consistent association between general phone use and self-reported maize yield” (in terms of the number of contacts, SMS sent and received, and calls made and received in short intervals). 

Based on these findings, authors proposed policy recommendations including support for intentional use of mobile phones for agricultural production and the relevance of farmers’ perception about mobile phone usage in agriculture. Their main conclusion is that there are important policy prospects that can help leverage current ICT usage to upsurge the extent of agricultural gains associated with mobile use in developing countries. 

General observation: The topic is nice and can be relevant in advancing an understanding of the constraints to adoption of agricultural technologies like ICTs and other critical technologies that contribute to the sustainable development goals (SDGs) especially hunger and climate change. However, the current version of the paper needs substantial improvement or a complete overhaul if it is to be acceptable for publication. Furthermore, authors should let a non-author read the paper for consistency checks before making any resubmission. 

RESPONSE: As noted above, we have made substantive revisions to every section of the paper to improve the quality of the writing and the clarity of our communication. We have made revisions to the modeling and presentation of data and findings, including overhauls of each of the tables and figures. We believe these revisions, which are described in greater detail throughout our response to reviewers’ comments, greatly improve the quality of the paper.

Major comments 

A. Writing, narrative, and structure 

1. The paper is not well written. It lacks a coherent structure, which makes it hard for me to see what the authors are really trying to achieve. 

RESPONSE: Substantive revisions to the quality of the writing have been made in each section of the manuscript.

2. I do not see a compelling contribution to literature. The potential contribution mentioned on pg. 5 (lines 111-120) has some problems including the endogeneity of perception due to its subjectivity, which the authors do not seem to address. 

- Authors do not clearly articulate the study’s relevance beyond the setting. 

- The proposed policy contributions on pg. 6 (lines 135-143) appear more like study objectives than actual policy contributions. Better writing can be helpful here. 

RESPONSE: not addressed

3. Insufficient motivation for use perception as the key policy variable in this study. Why should we care about the perception of farmers in the use of mobile phones for agriculture when the technology is not new? 

- What is novel about mobile phone utilization in agriculture in the region, elsewhere in 

Africa, and beyond? I would like to see a more convincing justification on this subject and the use of perception as a policy var. Perception, in fact, is a major determinant of technology adoption and/or program participation in many contexts. 

- I don’t see what is new about farmers’ perception of mobile phone use for agriculture in this context. 

RESPONSE: We see that the use of the word perception is confusing. In the paragraph of the introduction where we describe our specific study, we now define our use of the term “perceptions.”

4. Another critical problem is how agricultural productivity is measured – crop yields are not the only measure of productivity. 

- Self-reported yield estimates are weak due to several associated errors. 

- What actions did you take to account for this risk? See relevant example papers on this critical topic (Amadu et al., 2020; Desiere and Jolliffe, 2018; Godlonton et al., 2017; Judge and Schechter, 2009; Wossen et al., 2019). 

RESPONSE: not addressed

5. Insufficient description of the study area and data. Figure 1, the google earth map, is blurry. Please use an actual GIS map with coordinates to show your study sites (e.g., Coulibaly et al., 2017; Rana, and Miller, 2019; Van Campenhout, 2017). In general, all the Figures are blurry and unacceptable. 

RESPONSE: We apologize for the low resolution figures in the original submission. Regarding the map, it was not Google Earth imagery but an SRTM DEM surface to display elevation, overlaid with road networks and household locations. However following the reviewers advice, we include a revised, simplified study region map.

B. Data, conceptual framework and variables 

1. There is no conceptual framework for the paper. Authors can do a better job by presenting a conceptual framework that guides the statistical operations in the paper. This is critical for any quantitative analysis.

RESPONSE: not addressed

2. Therefore, the “Data analysis” section (pg. 10+) does not make much sense to me. For instance, how are the key variables measured like perception, various uses of mobile in relation to agriculture? See for example (Amadu et al., 2020; Rana, and Miller, 2019). 

RESPONSE: Substantial revisions have been made throughout the methods section.

3. Therefore, the two models presented on pg. 11 (lines 254 & 255) and not clear. 

RESPONSE: Substantial revisions have been made throughout the methods section.

4. Paper lacks a convincing presentation of data. I kept revising the paper to see: 

- how can I understand and interpret the data and results (albeit not really a result, but descriptive stats – more on this below). 

- For example, on pg. 9 (lines 193 – 196), I am confused as to whether some household members did not undertake farming? Is this articulated in the narrative? If yes, where? If no, why not? 

RESPONSE: We have made substantial improvements to our methods section and each of our tables.

5. Moreover, I suggest you provide a “theoretical expectations” section to guide the interpretation of the key variables based on sound literature review. This is lacking and need attention. 

- For example, you may want to caption part of pg. 12 to 13 as either literature review, or theoretical expectation, instead of making statements like “these covariates were informed by literature...” (pg. 13, L. 286). 

- We expect literature to inform your work. Therefore, just present a section to discuss the main vars and covariates in terms of what obtains elsewhere and expectations in this setting. 

RESPONSE: While we avoid creating a new section, we have added justifications for our choice of variables in appropriate sections.

6. Following the above points, there should been a summary stats’ table upfront to show the mean of all variables in the analysis. 

RESPONSE: Table 2 now does this for the regression models.

7. On pg. 9, writing on lines 198 to 206 is wordy. Consider using concise language for 

clarity. For e.g., use pretesting of questionnaires, instead of “conducted a practice survey ...” (lines 202-203). On the same pg., (lines 209-212), delete sentence starting with “All respondents ... study area”. It is wordy and adds little. 

RESPONSE: This language has been revised.

C. Results, discussion, policy, limitation, and conclusion 

1. The presentation of your results seems clumsy. For instance, your “results” (pg. 14) look 

more like descriptive statistics. Thus, I think Table 1 should be labelled ‘summary statistics’ rather than “results based on ...”. These are not rigorous analytical results. 

RESPONSE: We’ve made revisions to each table and split table 1 into 2 tables. Each table has a revised title. And summary statistics for model variables are not reported in table 2. 

2. Some variables in Table 1 had not been discussed in the narrative of the paper – the more reason you need to have a theoretical expectations section above. For example, “good year” is mentioned on pg. 11 (line 241) but not defined. “Bad year” is not mentioned earlier at all. Likewise, “synthetic phone” had not been defined earlier in the narrative, except for the mention of “synthetic variables” on pg. 12 (line 271). 

RESPONSE: Greater care has been taken with our descriptions/representations of all variables in the text and in the tables.

3. Table 1 and all Tables should have notes immediately below, not above in the Table title. 

RESPONSE: We have moved the notes below the table.

4. Table 2 does not have a good title. Consider presenting the title of a Table as a statement 

like ‘proportion of respondents using phones.” Moreover, there should be a note under the Table to provide clarity. The table is not clear. Is there a column for the interaction of these variables such as using phones to discuss with friends and for selling crops? The Table does not present a complete picture. 

RESPONSE: All the tables have new titles.

5. Table 3 is not clear. Same points as above such as proper labelling and notes. 

RESPONSE: We have also adjusted this table in accordance with these notes. 

6. The “discussion” section (pg. 20 – 22) reads more like a results section. 

RESPONSE: Substantial revisions have been made to the discussion section so that it better compares our findings to the findings from other studies.

7. On pg. 24, “intensity and frequency” appear for the first time in the paper. Why? 

- These are loaded terms, which should have been described in introduction or study area description before using them anywhere else. 

- Or they should have been conceptualized/operationalized in this study and then included in the analysis before using them here in the “limitations...” section. \\

RESPONSE: We recognize that these words are loaded in some disciplines and have thus changed them to more appropriate words for this context. 

8. I like your discussion of potential mechanisms (pg. 21) through which mobile phones 

may lead to yield enhancement. However, you do not provide statistical analyses for any of these mechanisms in the paper to bolster your findings (see. For example, Van Campenhout, 2017). 

RESPONSE: Throughout the manuscript now, we avoid the word “mechanism/s” except when discussing alternative approaches to examining the effect of mobile phones on agricultural productivity. 

9. Following the above point, there should be a robustness check for your findings. 

RESPONSE: not addressed

10. The conclusion is too terse and makes no sense. Should be succinct but convey enough 

information to appear as ‘stand-alone’ for an impatient reader. For example, the starting sentence (pg. 26 line 518) is awkward. In short, I would not consider this section as a conclusion for the paper because it presents little or nothing about the rest of the paper. 

RESPONSE: We’ve revised the conclusion so that it is summative and forward looking without replicating the abstract.

Other/Minor comments 

1. Be consistent about the use of sub-Saharan Africa versus Africa. To maintain the flow, 

choose one and stick to it, or you can indicate your intentional inter-use at the beginning of the paper and move on. 

RESPONSE: We’ve changed everything to sub-Saharan Africa.

2. Introduction is too long being 4 pages. Reduce to 2 or 2.5 pages. 

RESPONSE: The introduction is slightly shorter now.

3. On pg. 5 (line 108) remove many before fewer studies. 

RESPONSE: Done.

4. On pg. 6 (line 140-143), delete sentence starting with “Thus... and ending with 

productivity”. That whole sentence is not only poorly written but does not really add much to the argument of the paper. 

RESPONSE: This sentence has been shortened and rewritten.

5. On pg. 7 (L. 156) insert ‘distance from’ after “in relation to” ... 

RESPONSE: Done.

6. Also, on pg. 7, delete from lines 167 – 168. It adds nothing. 

RESPONSE: Done.

7. On pg. 8 (L. 171), write Data (delete collection). On line 173, write ‘Data for this study 

come from before “Focus group...” 

RESPONSE: The beginning of this paragraph has been adjusted.

8. Pg. 9: On L. 193, delete “both a”. On L. 194, write members. 

RESPONSE: Done.

9. On pg. 12, line 266 is missing something. Following line 255, I expected the two models. 

RESPONSE: This whole section has been revised. 

10. The entire paper can be significantly enhanced if the authors can clean up typos. 

RESPONSE: Substantial revisions have been made in each section. And we’ve done our best to root out all typos and grammatical errors.

References 

Amadu, F. O., Miller, D. C., McNamara, P. E., 2020. Yield effects of climate-smart agriculture aid investments in southern Malawi. Food Policy. 92, 101869. https://authors.elsevier.com/a/1ay3A15oGp6Skf. 

Coulibaly, J.Y., Chiputwa, B., Nakelse, T. and Kundhlande, G., 2017. Adoption of agroforestry and its impact on household food security among farmers in Malawi. Agricultural Systems, 155, 52-69. 

Desiere, S., Jolliffe, D., 2018. Land productivity and plot size: Is measurement error driving the inverse relationship? Journal of Development Economics. 130, 84–98. https://doi.org/10.1016/j.jdeveco.2017.10.002. 

Godlonton, S., Hernandez, M. A., Murphy, M., 2017. Anchoring Bias in recall data: Evidence from Central America. Amer. J. Agr. Econ. 0(0), 1–23; doi: 10.1093/ajae/aax080. 

Judge, G., Schechter, L., 2009. Detecting Problems in Survey Data Using Benford's Law. Journal of Human Resources 44 (1) 1-24. 

Rana, P., Miller, D. C., 2019. Explaining longterm outcome trajectories in social–ecological systems. PLoS ONE 14(4): e0215230. https://doi.org/10.1371/journal.pone.0215230. 

Van Campenhout, Bjorn., 2017. There is an app for that? The impact of community knowledge workers in Uganda, Information, Communication & Society, 20:4, 530-550, DOI: 10.1080/1369118X.2016.1200644 

Wossen, T., Alene, A., Abdoulaye, T., Feleke, S., Manyong, V., 2019. Agricultural technology adoption and household welfare: Measurement and evidence. Food Policy 87, 101742. https://doi.org/10.1016/j.foodpol.2019.101742.

---

## [Editor Report · Decision Letter 1]

24 Jul 2020

Mobile phone use is associated with higher smallholder agricultural productivity in Tanzania, East Africa

PONE-D-20-05740R1

Dear Dr. Quandt,

After reading your responses to earlier comments, I have decided that your manuscript has been judged scientifically suitable for publication and will be formally accepted for publication once it meets all outstanding technical requirements.

Kind regards,

Bjorn Van Campenhout, Ph.D.

Academic Editor

PLOS ONE
---

## [Editor Report · Acceptance letter]

29 Jul 2020

PONE-D-20-05740R1 

Mobile phone use is associated with higher smallholder agricultural productivity in Tanzania, East Africa 

Dear Dr. Quandt:

I'm pleased to inform you that your manuscript has been deemed suitable for publication in PLOS ONE. Congratulations! Your manuscript is now with our production department. 

Kind regards, 

on behalf of

Dr. Bjorn Van Campenhout 

Academic Editor

PLOS ONE